# ANT-Mediated Inhibition of the Permeability Transition Pore Alleviates Palmitate-Induced Mitochondrial Dysfunction and Lipotoxicity

**DOI:** 10.3390/biom14091159

**Published:** 2024-09-15

**Authors:** Natalia V. Belosludtseva, Anna I. Ilzorkina, Dmitriy A. Serov, Mikhail V. Dubinin, Eugeny Yu. Talanov, Maxim N. Karagyaur, Alexandra L. Primak, Jiankang Liu, Konstantin N. Belosludtsev

**Affiliations:** 1Institute of Theoretical and Experimental Biophysics, Russian Academy of Sciences, Institutskaya 3, 142290 Pushchino, Russia; 2Department of Biochemistry, Cell Biology and Microbiology, Mari State University, pl. Lenina 1, 424001 Yoshkar-Ola, Russia; dubinin1989@gmail.com (M.V.D.); bekonik@gmail.com (K.N.B.); 3Prokhorov General Physics Institute of the Russian Academy of Sciences, Vavilov St. 38, 119991 Moscow, Russia; 4Federal Research Center “Pushchino Scientific Center for Biological Research of the Russian Academy of Sciences”, Institute of Cell Biophysics of the Russian Academy of Sciences, Institutskaya 3, 142290 Pushchino, Russia; 5Medical Research and Education Institute, Lomonosov Moscow State University, 27/1, Lomonosovsky Ave., 119191 Moscow, Russia; 6School of Health and Life Sciences, University of Health and Rehabilitation Sciences, Qingdao 266071, China; j.liu@mail.xjtu.edu.cn

**Keywords:** lipotoxity, palmitate, mitochondria, adenylate translocator, bongkrekic acid, carboxyatractyloside, MPT pore, oxidative stress

## Abstract

Hyperlipidemia is a major risk factor for vascular lesions in diabetes mellitus and other metabolic disorders, although its basis remains poorly understood. One of the key pathogenetic events in this condition is mitochondrial dysfunction associated with the opening of the mitochondrial permeability transition (MPT) pore, a drop in the membrane potential, and ROS overproduction. Here, we investigated the effects of bongkrekic acid and carboxyatractyloside, a potent blocker and activator of the MPT pore opening, respectively, acting through direct interaction with the adenine nucleotide translocator, on the progression of mitochondrial dysfunction in mouse primary lung endothelial cells exposed to elevated levels of palmitic acid. Palmitate treatment (0.75 mM palmitate/BSA for 6 days) resulted in an 80% decrease in the viability index of endothelial cells, which was accompanied by mitochondrial depolarization, ROS hyperproduction, and increased colocalization of mitochondria with lysosomes. Bongkrekic acid (25 µM) attenuated palmitate-induced lipotoxicity and all the signs of mitochondrial damage, including increased spontaneous formation of the MPT pore. In contrast, carboxyatractyloside (10 μM) stimulated cell death and failed to prevent the progression of mitochondrial dysfunction under hyperlipidemic stress conditions. Silencing of gene expression of the predominate isoform ANT2, similar to the action of carboxyatractyloside, led to increased ROS generation and cell death under conditions of palmitate-induced lipotoxicity in a stably transfected HEK293T cell line. Altogether, these results suggest that targeted manipulation of the permeability transition pore through inhibition of ANT may represent an alternative approach to alleviate mitochondrial dysfunction and cell death in cell culture models of fatty acid overload.

## 1. Introduction

The availability of glucose and lipids in abundance, which is now widespread in the diet in many areas of the world, is a major cause of the ever-increasing incidence of obesity, insulin resistance, diabetes mellitus, and other metabolic disorders. High glucose concentrations (i.e., hyperglycemia) and a high content of long-chain saturated free fatty acids (i.e., hyperlipidemia) lead to a situation of severe metabolic stress in many cell types, especially in the vascular endothelium [1,2]. Notably, hyperlipidemia is a major risk factor for vascular lesions in diabetes and other metabolic disorders, although its basis remains poorly understood.

At the intracellular level, obesity-related diabetes and hyperlipidemia are known to be associated with the development of oxidative stress and mitochondrial damage [3,4]. Several studies using in vitro and in vivo models of diabetes and its comorbidities have demonstrated increased ROS production by the electron transport chain, impaired oxidative phosphorylation, a drop in the mitochondrial membrane potential, and the opening of a non-selective megachannel called the mitochondrial permeability transition (MPT) pore [4]. Furthermore, ultrastructural abnormalities in these organelles and damage to the mitochondrial quality control system, including the pathways of mitophagy, mitochondrial biogenesis, and dynamics, have been observed [4,5,6].

Accumulating evidence suggests that a decrease in ROS generation by mitochondria, restoration of mitochondrial quality control and function, as well as suppression of proapoptotic mitochondrial events are promising therapeutic strategies for diabetes and metabolic disorders. Mitochondria-targeted antioxidants have been shown to mitigate the negative impacts of obesity [7,8]. Modulators of AMP-activated protein kinase (AMPK), peroxisome proliferator-activated receptor gamma coactivator 1-alpha (PGC1α), dynamin-related guanosine triphosphatase (Drp1), and a number of other proteins that can regulate mitochondrial morphology and function have a protective effect against diabetes and its related complications [4,9,10,11]. Some studies have revealed that the specific blockers of the MPT pore opening, cyclosporin A and alisporivir, can attenuate cell damage in cell cultures exposed to hyperglycemia, as well as reduce mitochondrial abnormalities and overall manifestations of diabetes in a high-fat-diet mouse model [4,12,13].

The MPT pore is a Ca^2+^-dependent megachannel consisting of outer and inner mitochondrial membrane proteins, whose formation results in membrane permeabilization to 1.5-kDa molecules. This leads to disruption of ion homeostasis across the inner membrane, collapse of the membrane potential, swelling or destruction of mitochondria, and subsequent initiation of apoptotic or necrotic cell death [14,15,16,17]. Frequent and extended opening of the MTP pore has been shown to go alone with bursts of mitochondrial ROS and more persistent and cumulative oxidative damage to postmitotic cells [18,19]. The key regulator of the MPT pore is cyclophilin D, the target of cyclosporin A and its derivatives. The main component of the pore in the outer membrane is the voltage-dependent anion channel (VDAC). The adenine nucleotide translocator (ANT), along with ATP synthase, are considered to be the most important proteins that form the pore channel in the inner mitochondrial membrane [14,15,16,20].

ANT is known to be a protein that carries out the exchange transport of ATP to ADP and exhibits an intrinsic uncoupling activity in mitochondria. On the one hand, a decrease in the activity or knockout of this protein can be accompanied by a switch in energy metabolism to glycolysis [21,22]. Recent studies have shown that ANT depletion in kidney cells leads to metabolic reprogramming and preservation of overall renal health in a mouse model of obesity-induced chronic kidney disease [23]. On the other hand, ANT has been proposed to be a structural component of the MPT pore, a strategic regulator of cell death [4,14,15,24]. Therefore, the precise mechanism underlying the cytoprotective effect of modulation of this protein requires a more detailed consideration.

As is known, the mitochondria-targeted agents bongkrekic acid (BA) and carboxyatractyloside (CAT) are highly-affinity modulators of the ANT of the inner mitochondrial membrane [21,25,26,27]. Both of the ANT inhibitors similarly inhibit ADP-stimulated respiration by blocking the transport of adenine nucleotides across the membrane. However, BA is able to block the MPT pore, whereas CAT favors the MPT pore opening in mitochondria [21]. The difference in effects of the two modulators is believed to be due to the fact that BA stabilizes the ANT protein in the “m” (matrix-facing) conformation and CAT in the “c” (cytoplasm-facing) conformation [28]. Some studies have suggested that the inhibition of the MPT pore opening by BA underlies its anti-apoptotic and cytoprotective effects in a number of cell models for drug-induced toxicity [29,30,31].

The purpose of this work was to study the effects of BA and CAT on the progression of mitochondrial dysfunction in a primary culture of mouse pulmonary vascular endothelium under conditions of hyperlipidemia. The vascular endothelium is one of the first to suffer from such metabolic stress. Damage to the endothelium in micro and macrovessels is considered not only to be an early disorder of the cardiovascular system but also the leading cause of death in individuals with diabetes [32,33,34,35,36]. We assessed the action of bongkrekate and CAT on cell survival, ROS production, mitochondrial depolarization, the MPT pore opening, and colocalization of mitochondria with lysosomes in endothelial cells exposed to palmitate-induced lipotoxicity (0.75 mM palmitic acid/BSA for 48 h). In parallel, we investigated the effect of ANT2 silencing on the development of oxidative stress, mitochondrial damage, and viability of stably transfected HEK293T cells exposed to hyperlipidemic conditions.

## 2. Materials and Methods

### 2.1. Cell Cultures

Mouse (male BALB/c, 20–22 g) lung microvessels were used to isolate endothelial cells by indirect magnetic separation using rabbit polyclonal anti-CD31-antibodies (cat.# ab124432; Abcam, Cambridge, UK), as well as magnetic beads conjugated with goat anti-IgG-antibodies (cat.#11203D; Thermo Fisher, Waltham, MA, USA) [37]. Cells were cultured following standard protocol. For this purpose, DMEM: F12 medium (1:1) was used, additionally containing 10% fetal bovine serum, 100 U/mL penicillin, 2 mM L-glutamine, 50 μg/mL streptomycin, and 50 µg/mL endothelial cell growth supplement from bovine neutral tissue (ECGS, cat. # E2759; Sigma-Aldrich, St. Louis, MO, USA). The culture contained cells obtained from three animals. Cells of passages 7–10 were used in all experiments. Round coverslips (25 mm diameter) were placed one at a time into the wells of 6-well plates. A 0.2% gelatin solution was applied to the coverslips, then successively dried and a suspension of endothelial cells in the culture medium was added. Cells were cultured for 3 days until 90% confluence or greater was achieved.

HEK293T cells were obtained from the biobank of the Institute for Regenerative Medicine, Lomonosov Moscow State University, collection ID: MSU_HEK293 (https://human.depo.msu.ru, accessed on 14 September 2024). Cells were cultured according to standard protocol. For this purpose, we used DMEM medium additionally containing 10% fetal bovine serum and 2 mM L-glutamine without the addition of antibiotics. Cells from passages 10–13 were used in all experiments. For the experiment, cells were seeded into a 12-well plate at a density of 150 thousand cells/well. Cells were cultured for 2 days until 80–90% confluence was achieved.

### 2.2. CRISPR/Cas9-Mediated Knockdown of the SLC25A5 Gene in HEK293T Cells

A pair of pSpCas9(BB)-2A-GFP (PX458) (Addgene, Watertown, MA, USA #48138) constructs, modified to express Cas9-D10A nickase variant, was used to knockout *SLC25A5* gene (NC_000023.11 Chromosome X Reference GRCh38.p14 Primary Assembly: *SLC25A5*, Gene ID: 292, Nucleotide ID NM_001152.5, https://www.ncbi.nlm.nih.gov/gene/292, accessed on 14 September 2024) in HEK293T cell line. gRNA protospacers were designed and cloned in pSpCas9(BB)-2A-GFP_nickase using BbsI sites as described earlier [38,39]. The specificity of gRNA protospacers was assessed using COSMID: CRISPR Search with Mismatches, Insertions, and/or Deletions [40]. Obtained protospacers are listed in Table 1. HEK293T cell cultures were transfected with the obtained plasmid constructs as described earlier [41], and the 10% of FAM-brightest cells were sorted using a BD FACS Aria III cell sorter 72 h after transfection. In one-two weeks after the first transfection-sorting cycle, HEK293T cultures were re-transfected, and the 10% of FAM-brightest cells were sorted again in order to increase editing efficiency [42]. The resulting cell population was cloned.

To confirm *SLC25A5* knockdown, genomic DNA was isolated from the obtained clones and amplified using the primers listed in Table 1. Amplicons were sequenced by Sanger sequencing, and the results were analyzed using Chromas 2.6.6 software (Technelysium Pty Ltd., South Brisbane, Australia). The design and sequencing results of CRISPR/Cas9-mediated editing of the *SLC25A5* gene in HEK293T are demonstrated in Appendix A. The “TIDE: Tracking of Indels by DEcomposition” software (version 3.3.0) was used for analysis of sequencing results of the edited genes to elucidate the effectiveness of genome editing [43].

### 2.3. Modeling of Palmitate-Induced Lipotoxicity

Lipotoxicity was modeled by incubating endothelial cells in a culture medium containing 0.75 mM palmitic acid (PA) (complexed with fatty acid-free bovine serum albumin (BSA, cat. # A6003; Sigma-Aldrich, St. Louis, MO, USA)) (0.5 mM palmitic acid for HEK293T) for 2 or 6 days in a CO_2_ incubator (Sanyo, Osaka, Japan). Control cells were treated for the same time with fatty acid-free BSA alone. The preparation of complexes of palmitic acid/BSA was carried out as shown in [44].

Bongkrekic acid (BA, cat. # 203671; Merck KGaA, Darmstadt, Germany) and carboxyatractyloside (CAT, cat. # PHL84196; Merck KGaA, Darmstadt, Germany) were applied to cell cultures in the form of a dimethyl sulfoxide (DMSO) solution. The inhibitors BA and CAT were added simultaneously with the palmitic acid/BSA complex to the respective experimental groups, followed by medium replacement every 48 h, maintaining the concentration of all compounds at a constant level throughout the experiment (2 or 6 days). In parallel, additional control groups of cells were incubated for the same time in the medium containing a DMSO solution. The total volume of DMSO was always less than 0.1%, which had a negligible effect in cell culture experiments.

### 2.4. Cell Viability Assessment

Cell viability was assessed after 6 days of incubation in the presence of palmitate (0.75 mM) using an EVOS FLoid Imaging System (Thermo Fisher, Waltham, MA, USA). For this purpose, the cells were washed three times with Hank’s solution and then incubated with 5 μg/mL Hoechst 33,342 vital dye and 5 μM propidium iodide at 37 °C. Data were collected from at least 25 fields of view for each experimental condition. Images were analyzed using Image J2 (Fiji) software (National Institutes of Health, Bethesda, MD, USA) [12].

### 2.5. Mitochondrial Function Assay

To determine the mitochondrial potential, cells were stained with TMRM (tetramethylrhodamine methyl ester) at 10 nM for 30 min at 37 °C [45]. Next, the cells were washed with PBS and analyzed using a fluorescence microscope as described previously. Fluorescence was recorded at excitation/emission wavelengths of 550/575 nm. Fluorescence was assessed using an AE31E inverted microscope (Motic, Barselona, Spain) equipped with a Motic PLAN FLUAR 10× N.A. 0.3 objective. Light source brightness, exposure time, and gain settings were the same throughout all experiments. After recording the initial intensity of the fluorescent signal, depolarization of mitochondria was induced using 2 μM FCCP (carbonyl cyanide p-trifluoro methoxyphenylhydrazone), and the dynamics of the decrease in fluorescence intensity was recorded (2 s/frame) until the moment when the fluorescence intensity stopped changing. The mitochondrial potential was calculated as the ratio of the difference between the maximum and minimum fluorescence intensities to the maximum fluorescence intensity and expressed as a percentage. Background fluorescence levels were subtracted from all values.

Mitochondrial pore opening was assessed using calcein-AM fluorescence in the presence of 1 mM CoCl_2_ [12,46]. After completing incubation with the tested agents, the cells were washed three times with Hank’s solution and incubated for 30 min at 37 °C in the presence of 1 μM calcein-AM, 200 nM MitoTracker Red (to visualize mitochondrial structure), and 1 mM CoCl_2_. After the staining procedure, cells were washed with Hank’s balanced salt solution, and fluorescence was assessed using a DMI6000 confocal microscope (Leica Microsystems, Wetzlar, Germany).

ROS production was detected using the fluorescent dye 2’,7’-dichlorodihydrofluorescein diacetate (H_2_DCFDA). Cells were stained with 20 μM H_2_DCFDA for 30 min at 37 °C [12]. Cell fluorescence was assessed using an EVOS FLoid Imaging System (Thermo Fisher, Waltham, MA, USA).

The concentration of superoxide anion radical in mitochondria was measured using MitoSOX Red (Thermo Fisher, USA) according to the protocol recommended by the manufacturer. After incubation, cells cultured on round coverslips (25 mm diameter) were washed from the culture medium and 1 mL of complete Hanks solution with 1 µM MitoSOX Red was added and incubated for 30 min at 5% CO_2_ and 37 °C, then washed with 1 mL Hanks solution, transferred to a round coverslip chamber (Warner Instruments, Hamden, CT, USA), and 500 μL of complete Hanks solution was added. All solutions were preheated to 37 °C. Fluorescence recording was performed at wavelengths Ex/Em 535/620 nm. Fluorescence was assessed using an AE31E inverted microscope (Motic, Barselona, Spain) equipped with a Motic PLAN FLUAR 10× N.A. 0.3 objective. Light source brightness, exposure time, and gain settings were the same throughout all experiments.

Colocalization of mitochondria and lysosomes in endothelial cells was assessed using confocal microscopy. The method is based on the colocalization of two fluorescent dyes, MitoTracker DeepRed FM (200 nM) and LysoTracker Green (50 nM), in cells [12]. A DMI6000 microscope (Leica Microsystems, Wetzlar, Germany) was used to obtain confocal images. Colocalization was analyzed using Image J2 software (NIH, Bethesda, MD, USA). The proportion of the area of colocalization of mitochondria and lysosomes was estimated as a fraction of the total area of mitochondria in the field (taken as 100%). For each condition, four biological replicates were performed (cover slip with cells from a separate culture).

Images were analyzed using Image J2 (Fiji) software (National Institutes of Health, Bethesda, MD, USA).

### 2.6. Electrophoresis and Immunoblotting

Cells were washed twice with cold PBS and then treated with 2× loading buffer containing 4% SDS, 20% glycerol, and 100 mM Tris-HCl (pH 6.8). The resulting suspension was incubated for 10 min at 95 °C, resuspended by pipetting, and centrifuged at 10,000× *g* for 5 min. The supernatant was transferred to a new Eppendorf tube and used to determine the protein concentration using the Quick Start™ Bradford Protein Assay Kit. The samples were diluted in Laemmli buffer, run on 12.5% SDS-PAGE (10 µg of cell lysates per lane), and transferred to a 0.45 μm nitrocellulose membrane (Cytiva, Marlborough, MA, USA). After overnight blocking, the membrane was incubated with the appropriate primary antibody. The anti-ANT2 (A15639) and anti-GAPDH (AC001) were from Abclonal (ABclonal Germany GmbH, Düsseldorf, Germany). Chemiluminescent ECL reagents (Pierce, Rockford, IL, USA) were used to determine the peroxidase activity. Proteins were visualized using the LI-COR system (LI-COR, Lincoln, NE, USA). Optical density measurements were performed by LI-COR Image Studio software version 5.2.

### 2.7. Statistical Data Processing

Statistical data analysis was performed using GraphPad Prism version 8.4 (GraphPad Software Inc., San Diego, CA, USA). One-way analysis of variance (ANOVA) followed by Tukey’s *post hoc* test was used to evaluate the statistical significance of differences between experimental groups. Results were presented as the mean ± SEM from 3–5 independent experiments with different cell cultures.

## 3. Results

### 3.1. The Effect of Bongkrekic Acid and Carboxyatractyloside on the Viability of Mouse Lung Endothelial Cells under Conditions of Normal and Dyslipidemia

Figure 1A shows data on the viability index of mouse endothelial cells after 48 h treatment with BA or CAT at different concentrations under control (normolipidemic) conditions. One can see that the addition of 10–50 μM BA or 5–10 μM CAT had no significant effect on the viability of endothelial cells. Further increase in the concentration of CAT (25 μM) resulted in a slight (about 5%) increase in the proportion of dead cells under normal conditions. Therefore, further experiments were carried out with 25 μM BA or 10 μM CAT.

Then, we examined the effects of these drugs under conditions of chronic exposure of mouse endothelial cells to high concentrations of palmitic acid in the form of complexes with BSA, which models lipotoxic stress-induced cell injury. As is known, hyperlipidemia (chronic elevation of circulating free fatty acids) is a condition accompanied by significant disruption of the function of endothelial cells and resulting in cell death [7].

Figure 1B demonstrates that palmitic acid (0.75 mM) caused significant cell death, and the number of surviving cells decreased by almost 80% after 6 days of incubation (in the control group (CTR), the cell viability index (%) was 96.0 ± 3.0, as in Figure 1A). Pre-treatment of cells exposed to these conditions with 25 μM BA led to a significant increase in this index (at the end of the experiment, the number of living cells was 31%). At the same time, 10 µM CAT reduced the survival of endothelial cells under dyslipidemia conditions.

It should be noted that bongkrekate and CAT at micromolar concentrations were able to suppress ADP-stimulated mitochondrial respiration (Appendix A). Moreover, mitochondrial susceptibility to Ca^2+^-induced MPT pore opening, as measured by the calcium retention capacity index, was decreased upon the modulation of the ANT by BA, whereas its inhibition by CAT increased this index in isolated mitochondria (Appendix A).

### 3.2. Effect of Bongkrekic Acid and Carboxyatractyloside on the Development of Oxidative Stress in PA-Induced Lipotoxity

PA-induced lipotoxity has been proposed to trigger a number of aberrant pathways associated with oxidative stress and mitochondrial impairment [4]. So, we explored whether bongkrekate and CAT could affect mitochondrial dysfunction and ROS production in mouse lung endothelial cells in hyperlipidemic stress. Since 0.75 mM palmitic acid caused a sharp decline in cell viability within 6 days, in the next series of experiments we simulated lipotoxicity over a 48 h period. Our additional experiments showed that after this treatment regimen, the cell viability index was 71.0 ± 2.0%.

Figure 2A shows that incubation of endothelial cells with 0.75 mM palmitic acid for 48 h resulted in a significant (1.6-fold) increase in DCF fluorescence, suggesting the development of oxidative stress due to overproduction of ROS. The increase in DCF fluorescence intensity was less pronounced in the presence of bongkrekate (1.2-fold). One can see that there was a tendency for DCF fluorescence to decrease in the PA + BA group compared with the PA group. However, the change in this parameter was not statistically significant. In the presence of 10 µM CAT, a significant increase in DCF fluorescence was observed both in normo- and hyperlipidemia (1.7- and 1.9-fold, respectively). The level of DCF fluorescence in the CTR + CAT and PA + CAT groups did not differ from that in the PA group.

Figure 2B shows the changes in mitochondrial generation of ROS, mainly superoxide anion (O_2_^•−^), using the fluorescent probe MitoSOX Red. One can see that, similar to the case with DCF, there was a significant increase in fluorescence intensity in the presence of PA. These results suggest that mitochondria are the major source of ROS generation in endothelial cells under lipotoxic conditions. It should be noted that additional experiments with antimycin A, an inhibitor of complex III of the respiratory chain, demonstrated that the maximum level of ROS generation by mitochondria in the presence of palmitic acid was not achieved (the CTR + AA group, Figure 2B).

MitoSOX-based assays showed that in mitochondria, bongkrekate significantly suppressed PA-induced overproduction of ROS (the PA + BA group, Figure 2B). Interestingly, this effect of BA was not observed when determining total ROS production in the entire cell cytoplasm in experiments using the DCF probe (Figure 2A). Altogether, these data allow us to conclude that the main site of ROS formation under lipotoxic conditions is the mitochondrial respiratory chain. It is well known that under lipotoxic conditions, non-mitochondrial antioxidant systems can be damaged and/or cellular prooxidant systems (e.g., NADPH oxidases, endoplasmic reticulum, and peroxisomes) can also be activated, thereby increasing the overall production of ROS in the cell [47]. One can suggest that bongkrekate is unable to reduce the cytoplasmic pool of ROS.

In the presence of CAT, the fluorescence level of MitoSOX Red did not change significantly compared with that in the PA group. Moreover, CAT itself induced increased ROS generation in the cell cytoplasm (measured by DCF fluorescence intensity) but not in the mitochondria (measured by Mitosox Red fluorescence intensity). Apparently, CAT itself was able to further induce the production of ROS in cellular compartments.

### 3.3. Effect of Bongkrekic Acid and Carboxyatractyloside on the Development of Mitochondrial Dysfunction in PA-Induced Lipotoxity

PA-induced lipotoxity was accompanied by quenching of fluorescence of the mitochondrial membrane potential probe TMRM, indicating mitochondrial depolarization in mouse lung endothelial cells (Figure 3). The administration of 25 μM bongkrekate resulted in a significant restoration of the membrane potential of mitochondria in endothelial cells exposed to chronic elevation of palmitic acid. The addition of 10 µM CAT also partially restored the membrane potential (when compared with the PA group).

Bongkrekate and CAT are known not only as ligands for mitochondrial adenine nucleotide translocase but also as MPT-pore modulators [22,28]. As mentioned above, BA can inhibit the formation of the MPT pore in mitochondria, while CAT is a co-inducer of the MPT pore. We previously demonstrated that the formation of this pore is one of the pathological factors in the development of mitochondrial dysfunction in hyperglycemic conditions [12,46]. Here, we studied the effect of BA and CAT on the spontaneous opening of the MPT pore under PA-induced lipotoxity. Analysis of the MPT-pore opening was carried out by quenching the fluorescence of calcein loaded into mitochondria in the presence of cobalt ions. It is believed that when the integrity of the mitochondrial membrane is disrupted (due to the opening of the MPT-pore), CoCl_2_ penetrates into the mitochondria (or calcein may leave the organelles), which leads to quenching of calcein fluorescence.

Figure 4A shows typical patterns of calcein fluorescence in endothelial cells from four experimental groups stained with this indicator in the presence of cobalt ions.

One can see that the fluorescence intensity of calcein (in the presence of cobalt ions) in endothelial cells exposed to 0.75 mM palmitic acid (for 48 h) significantly decreased (Figure 4B). This suggests that the spontaneous opening activity of the MPT pore in endothelial cells increases under hyperlipidemic conditions. Incubation of endothelial cells with 25 μM BA under conditions of hyperlipidemia resulted in a significant increase in calcein fluorescence intensity (Figure 4). This indicates that BA can inhibit the MPT-pore opening in the inner mitochondrial membrane in endothelial cells under conditions of hyperlipidemic stress. Preincubation of endothelial cells with 10 μM CAT both under normal and hyperlipidemic conditions caused a decrease in calcein fluorescence compared to the CTR group. In parallel, there was also a tendency for this indicator to decrease compared to the PA group. This may be due to the fact that CAT itself causes spontaneous opening of the MPT pore.

Cell death resulting from the lipotoxic effects of palmitic acid may be associated with selective degradation of mitochondria [4]. In this regard, we assessed the effect of 25 μM BA and 10 μM CAT on the level of colocalization of mitochondria and lysosomes in mouse lung endothelial cells in hyperlipidemic stress. We assessed the degree of colocalization of these organelles using the method of double cell staining with the help of MitoTracker DeepRed FM for mitochondria (red) and LysoTracker Green for lysosomes (green) (Figure 5).

Figure 5 shows that under conditions of 48 h hyperlipidemia, the colocalization of mitochondria and lysosomes significantly increased, which may indicate increased mitochondrial elimination. In this case, the incubation of endothelial cells with 25 μM BA led to a decrease in the level of colocalization of mitochondria and lysosomes. On the contrary, 10 µM CAT did not have such an effect. In the PA + CAT group, the colocalization of lysosomes and mitochondria was higher than in the CTR and CTR + CAT groups, and it did not differ significantly from that in the PA group.

### 3.4. Suppression of ANT2 Expression Enhances the Development of Oxidative Stress and Death of HEK293T Cells under Conditions of Hyperlipidemia

As can be seen from previous studies, pharmacological inhibition of ANTs may lead to differential effects on mitochondrial dysfunction, oxidative stress, and cell viability under hyperlipidemia. It is known that ANT2 is the predominant isoform expressed in endotheliocytes, HEK293T cells, and hepatocytes. Reduction of the level of this isoform was shown to cause functional mitochondrial disorders and cell death [24].

To study the mechanisms of these phenomena, we constructed a HEK293T cell line with suppressed expression of the *SLC25A5* gene, which encodes the ANT2 protein. The obtained cell line (ANT2-HEK293T) demonstrated ~60% reduction of ANT2 protein expression according to Western Blot analysis (Figure 6A) and ~35% of DNA editing efficiency according to TIDE analysis of Sanger sequencing chromatograms (Appendix A). Using this cell culture, we studied the effect of reduced ANT2 expression on mitochondrial dysfunction, oxidative stress, and cell survival under conditions of PA-induced lipotoxicity.

Figure 6B shows that there was a statistically significant difference in the viability index of control HEK293T (WT) and ANT2-HEK293T cells. After 6 days of incubation in the presence of 0.5 mM palmitic acid, significant death of control cells was observed, and the number of surviving cells decreased by almost 2.5 times (the number of living cells was 37% of the control value). In the group of ANT2-HEK293T, cell death was also enhanced under palmitate supplementation, and the number of surviving cells was ~25% of the number of cells under normolipidemic conditions.

Figure 7A shows that incubation of HEK293T cells with 0.5 mM palmitic acid for 48 h resulted in a significant (1.4-fold) increase in the fluorescence level of DCF, suggesting the development of oxidative stress due to overproduction of ROS. HEK293T cells with reduced ANT2 expression showed a 1.6-fold increase in ROS production under these conditions. Moreover, ANT2 deletion caused an increase in DCF fluorescence even in the absence of palmitic acid.

Suppression of ANT2 expression in HEK293T cells also led to a decrease in mitochondrial membrane potential upon hyperlipidemia, but it was less pronounced compared to that in wild-type cells (Figure 7B).

In addition, we assessed the fluorescence intensity of calcein (in the presence of cobalt ions) in HEK293T cells with normal and reduced expression of ANT2 to study the spontaneous opening of the MPT pore in the mitochondria (Figure 8).

One can see that the mitochondria from HEK293T with normal and reduced expression of ANT2 were sensitive to spontaneous formation of the MPT pore.

## 4. Discussion

Dyslipidemia is among the key disorders that significantly contribute to the development of obesity, insulin resistance, and diabetes mellitus. Among the entire spectrum of pathogenetic factors, mitochondrial alterations and associated oxidative stress have turned out to be the predominant mechanisms of lipotoxicity [4]. As is shown in the present work, lipotoxicity caused by PA goes along with an additional increase in mitochondrial susceptibility to MTP pore formation, ROS overproduction, depolarization of the inner mitochondrial membrane, and intensification of the process of colocalization of lysosomes with the mitochondria in mouse primary lung endotheliocytes and HEK293T cells. These phenomena are exacerbated by cell death. It is noteworthy that mitochondrial dysfunction in hyperlipidemic stress can occur in various cell types, including cardiomyocytes, astrocytes, and cancer cells [48,49,50,51,52], but the vascular endothelium is one of the first tissues to face dramatic metabolic stress and mitochondrial damage in diabetes [32,33,34,35,36].

Increasing evidence suggests that targeting mitochondria may restore endothelial cell function in diabetes and exert antidiabetic effects. Mitochondria-targeted antioxidants, regulators of energy homeostasis, and the MPT pore formation demonstrated beneficial effects on mitochondrial capacity and function, insulin sensitivity, and diabetes-related cell damage [4,5,6,7,8,9,10,11,53,54,55]. Several recent studies have demonstrated the functional significance and therapeutic potential of mitochondrial membrane proteins with multiple physiological functions within the cell. In particular, pharmacological or genetic inactivation of the VDAC1 protein of the outer membrane mitigates mitochondrial dysfunction and oxidative stress in diabetes mellitus in in vitro and in vivo models [46,56,57,58]. VDAC1 is known to be a structure involved in the formation of the MPT pore and hetero-oligomeric channels with proapoptotic Bax/Bak proteins [59,60]. Moreover, inactivation of VDAC1, as the main pathway for the transport of metabolites across the membrane, can lead to changes in intracellular energy metabolism [59,61]. In this work, we tried to answer the question of whether it is possible to correct PA-induced lipotoxicity by regulating the inner membrane pore-forming component, ANT. On the one hand, ANT is a key structural component of the MPT pore, and its inhibition can prevent PA-induced apoptosis, suggesting a role for this channel in the development of cell death. On the other hand, ANT2 depletion in renal proximal tubule cells of model mice was found to trigger a cascade of metabolic reprogramming (a switch of metabolism from oxidative phosphorylation to glycolysis), which leads to enhanced renal cell survival and ultimately to the preservation of kidney function in obesity-induced chronic kidney disease [23].

Bongkrekate and CAT are highly specific inhibitors of the ANT protein [21,25,26,27]. Blocking ANT with BA or CAT can suppress mitochondrial transport of adenine nucleotides, thereby inhibiting the process of oxidative phosphorylation. In this case, the function of mitochondria as an ATP producer is limited [22]. However, the action of these drugs is manifested on the opposite sides of the inner mitochondrial membrane. Studies showed that BA stabilizes the ANT protein in the matrix (“m”) conformational state, which inhibits the formation of the MPT pore in mitochondria. On the contrary, CAT, by blocking ANT in the cytosolic (“c”) conformation state, is an inducer of the non-specific mitochondrial pore [28]. Thus, using these two drugs, it is possible to determine which process is directly related to the development of mitochondrial dysfunction during lipotoxicity—suppression of adenine nucleotide exchange or induction of the MPT pore.

As is shown in the present study, 25 μM BA can prevent the progression of main signs of mitochondrial dysfunction induced by hyperlipidemic stress in mouse endothelial cells. Indeed, this agent is able to alleviate PA-induced excessive mitochondrial ROS generation, increased mitochondrial susceptibility to spontaneous formation of the MPT pore, and mitochondrial depolarization in endothelial cells. Moreover, BA significantly suppresses the proportion of mitochondria colocalizing with lysosomes in cultured endothelial cells during dyslipidemia.

Increased colocalization of mitochondria and lysosomes, which is observed under conditions of hyperlipidemia, may indicate the degradation of mitochondria. One can suggest that this phenomenon is a consequence of the activation of processes leading to cell death or adaptive changes accompanied by the removal of damaged mitochondria through autophagy (mitophagy). At present, we have not investigated the exact mechanisms underlying the degradation of mitochondria during lipotoxicity. Some studies showed that palmitate can cause suppression of autophagy along with activation of apoptotic cell death [49]. Therefore, it can be assumed that BA is able to attenuate PA-induced pathological events in the cell and protect mitochondria from degradation.

An important effect of BA is an increase in cell survival in the presence of PA. Similar antiapoptotic and antinecrotic effects of BA have been demonstrated in various cell cultures in many studies [29,30]. It is believed that this is due to the suppression of MPT pore formation.

The effects of CAT on mitochondrial functions differ significantly from those observed in the presence of BA. In our experiments, CAT itself caused the induction of cell death and also stimulated cell death under conditions of PA-induced lipotoxicity. This was accompanied by an increase in the development of both oxidative stress (we observed this increase under conditions of normo- and hyperlipidemia) and mitochondrial dysfunction. Indeed, both under conditions of normo- and hyperlipidemia, spontaneous opening of the MPT pore was observed in the presence of CAT. It is important to note that the MPT pore can function in two functional modes—high- and low-conductivity states [20]. One could assume that in the presence of CAT, the mitochondrial pore opens in a low-conductivity state, whereas PA induces the formation of the MPT in a highly conductive state. This may explain why significant endothelial cell death occurs under lipotoxic conditions, but does not occur in the presence of CAT alone. Unfortunately, it is impossible to determine the mode of functioning of the pore based on the quenching of calcein in mitochondria.

Carboxytractyloside did not reduce the degree of colocalization of mitochondria and lysosomes. However, the PA + CAT group showed a restoration of mitochondrial membrane potential compared to the PA group. One may suggest that this is due to the blockade of electrogenic transport of ATP/ADP, which can lead to an increase in the membrane potential.

It should be noted that a decrease in ANT2 expression in HEK293T cells (the main isoform that ensures the transport of adenine nucleotides across the mitochondrial membrane in this cell culture, endothelial cells, and hepatocytes) leads to events that are similar to those obtained under the action of CAT. One can see an increase in the generation of ROS, a slight increase in the membrane potential, and no suppression of the MPT pore opening. This ultimately leads to increased cell death. According to the literature, ANT2-null embryos of mice die quite quickly, which is combined with cardiac developmental failure, immature cardiomyocytes having swollen mitochondria, cardiomyocyte hyperproliferation, and cardiac failure due to hypertrabeculation/noncompaction. It is believed that ANT2 biases the MTP toward closed, while ANT1 biases the MTP toward open [24]. It is therefore possible that even a partial reduction of ANT2 levels may lead to increased cell death through increased mitochondrial pore induction and ROS generation. However, it cannot be denied that in other cell cultures, there may be a different pattern, and the effect of diabetes on the mitochondrial pore opening may be tissue-specific.

Thus, BA has a protective effect, reducing cell death due to suppression of the development of oxidative stress and mitochondrial dysfunction. Contrariwise, a decrease in the level of ANT2 expression or the addition of CAT stimulates cell death under conditions of hyperlipidemia. The marker signal of this phenomenon is the increased generation of ROS in the cell. Taken together, the data obtained indicate that, at least in cell cultures, induction of the mitochondrial non-selective megachannel, the permeability transition pore, is a key event in mitochondrial dysfunction and lipotoxicity induced by palmitate. Changes in energy metabolism during lipotoxic stress, which may occur due to inhibition of the ANT protein, do not seem to play a prominent role in the progression of mitochondrial damage. Therefore, inhibition of the mitochondrial pore, rather than switching metabolism from oxidative phosphorylation to glycolysis, may be the mechanism that can save the cell from lipotoxicity. Furthermore, the work identifies ANT as a mitochondrial sensor for elevated fatty acids and an important target in hyperlipidemia-induced mitochondrial dysfunction and cell death of endotheliocytes and HEK293T cells in cell culture models of fatty acid overload.

## 5. Conclusions

In the current study, the causal role of ANT-mediated mitochondrial permeability transition in a primary culture of mouse pulmonary vascular endothelium and HEK293T cells exposed to hyperlipidemia has been critically examined. Our results suggest that opening of the MPT pore is a key event in the progression of PA-induced mitochondrial dysfunction, oxidative stress, and cell death in in vitro models of hyperlipidemia. The MPT pore activator (and ANT inhibitor) CAT has no protective effect against these pathological phenomena in a primary culture of mouse lung endothelial cells incubated with excessive amounts of palmitic acid. Similar to the effect of CAT, a decrease in ANT2 protein content in HEK293T cells can stimulate the development of oxidative stress both under conditions of normo- and hyperlipidemia. On the contrary, bongkrekate, an ANT and MPT pore blocker, is able to rescue endothelial cells from mitochondrial dysfunction and suppress cell death in hyperlipidemia. There is much evidence that this drug has antiapoptotic and antinecrotic effects in cell cultures under pathological conditions associated with oxidative stress. Thus, it seems that inhibition of the MPT pore rather than a metabolic switch from oxidative phosphorylation to glycolysis is the key event in protecting the cell from palmitate-induced lipotoxicity. However, it should be noted that BA and CAT are mitochondrial toxins that can inhibit the export of ATP molecules into the cytoplasm and cause death in healthy humans and animals [62]. Therefore, despite positive effects on cell cultures, caution must be exercised when using this agent for the treatment of mitochondrial disorders. One can assume that only mitohormesis doses of BA can be used as a tool for studying the mitochondria-mediated mechanisms of cell death in severe or chronic metabolic pathologies, including diabetes mellitus and its complications.

## Figures and Tables

**Figure 1 biomolecules-14-01159-f001:**
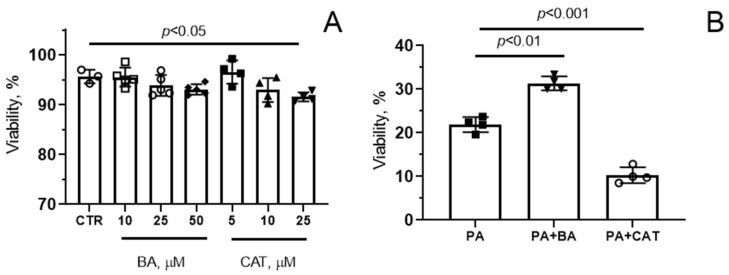
Effect of bongkrekic acid (BA) and carboxyatractyloside (CAT) on the viability of mouse lung endotheliocytes under conditions of normo- (**A**) and hyperlipidemia (**B**). (**A**) Cells were treated with BA and CAT at different concentrations for 48 h, and the cell viability index was quantified. (**B**) Effect of 25 µM BA and 10 µM CAT on palmitate (PA)-induced lipotoxicity (0.75 mM PA/fatty acid-free BSA complex solution for 6 days) in the mouse lung endothelial cells. Data represent the mean ± SD from 3–4 independent experiments, including at least 25 fields of view.

**Figure 2 biomolecules-14-01159-f002:**
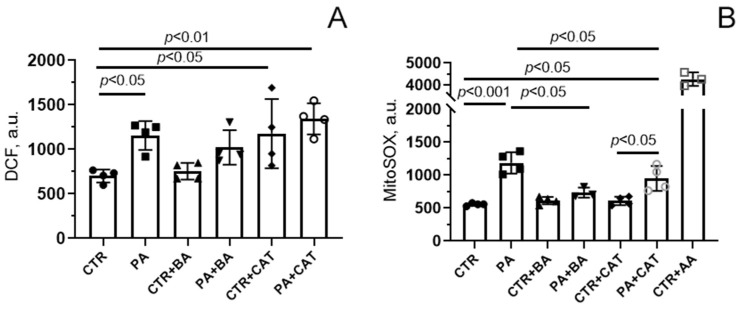
Effect of bongkrekic acid (BA, 25 µM) and carboxyatractyloside (CAT, 10 µM) on production of reactive oxygen species in mouse lung endothelial cells under conditions of palmitate lipotoxicity (0.75 mM PA/fatty acid-free BSA complex solution for 48 h). (**A**) DCF fluorescence level reflecting ROS production in the cell cytoplasm; (**B**) MitoSOX red fluorescence, reflecting the production of superoxide anion in the mitochondria. The addition of 10 μM antimycin A (AA), an inhibitor of the respiratory chain complex III, demonstrated the highest level of superoxide anion generation by mitochondria of mouse lung endothelial cells. Data represent the mean ± SD from 3–4 independent experiments, including at least 25 fields of view.

**Figure 3 biomolecules-14-01159-f003:**
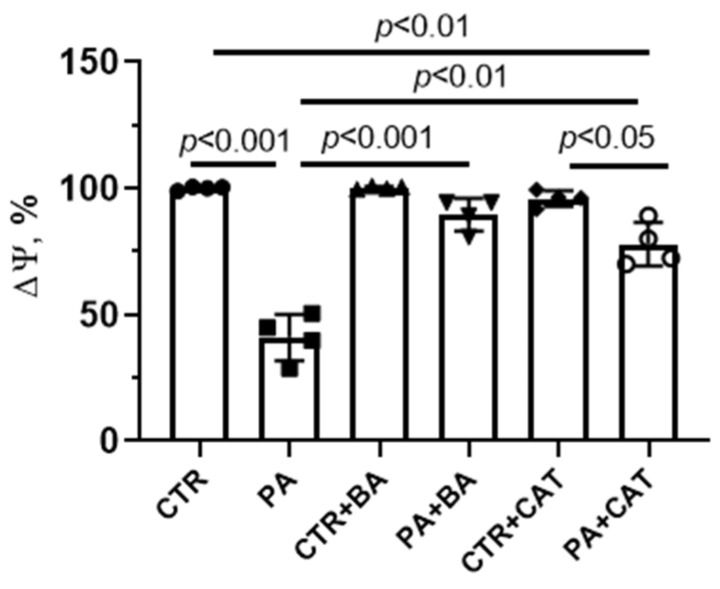
Effect of bongkrekic acid (BA, 25 µM) and carboxyatractyloside (CAT, 10 µM) on mitochondrial membrane potential (Δψ) in mouse lung endothelial cells under conditions of palmitate-induced lipotoxicity (0.75 mM PA/fatty acid-free BSA complex solution for 48 h). Data represent the mean ± SD from four independent experiments.

**Figure 4 biomolecules-14-01159-f004:**
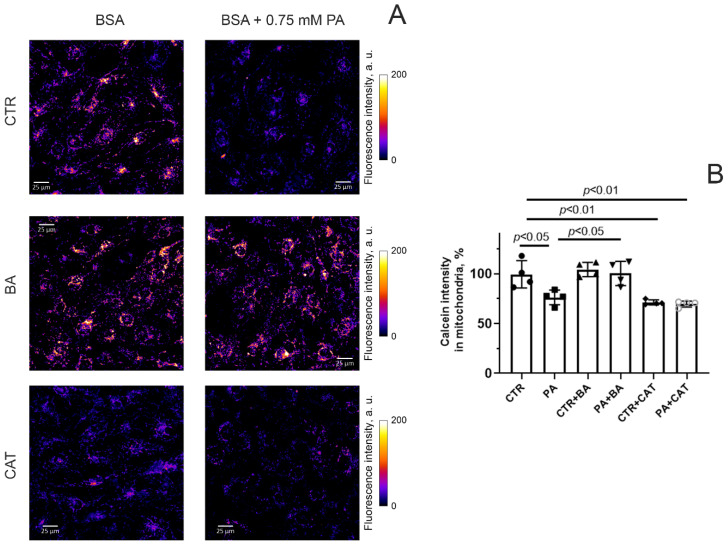
MPT pore opening in mouse lung endothelial cells. (**A**) Typical images of calcein fluorescence in the presence of CoCl_2_ in endothelial cells of the experimental groups. Scale bar—25 μm. (**B**) Intensity of calcein fluorescence in mitochondria of the mouse lung endothelial cells from six experimental groups. Conditions: CTR—BSA solution, PA—0.75 mM palmitate/BSA complex solution. Data represent the mean ± SD from four independent experiments, including at least 25 fields of view.

**Figure 5 biomolecules-14-01159-f005:**
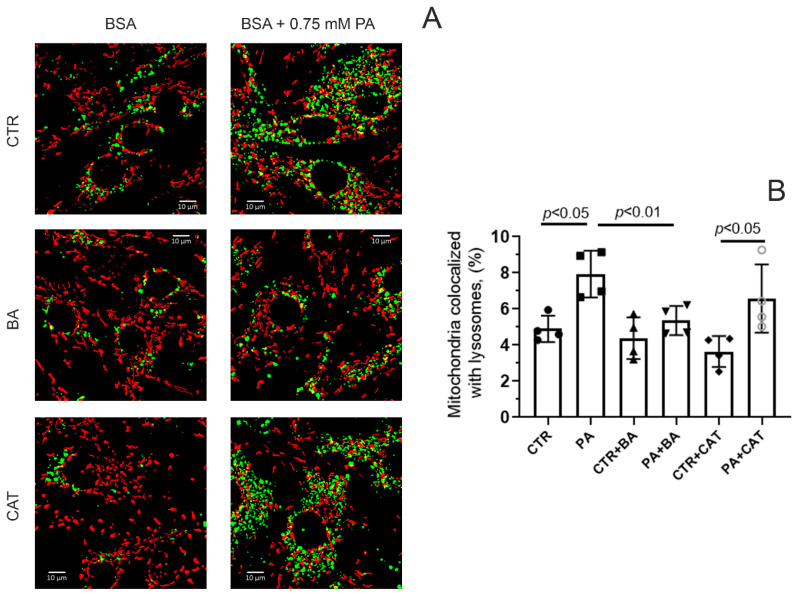
Effect of bongkrekic acid (25 µM) and carboxyatractyloside (10 µM) on the level of colocalization of mitochondria and lysosomes in endothelial cells during palmitate-induced lipotoxicity. (**A**) Typical fluorescence images of MitoTracker DeepRed FM (red dots) and LysoTracker Green (green dots) and their colocalization are shown. Scale bar—10 μm. (**B**) Number of mitochondria (%) colocalized with lysosomes in the mouse lung endotheliocytes from four experimental groups. Abbreviations used: BA, bongkrekic acid; CAT, carboxyatractyloside; PA, palmitic acid. Conditions: CTR—BSA solution, PA—0.75 mM palmitate/BSA complex solution. Data represent the mean ± SD from four independent experiments, including at least 10 fields of view.

**Figure 6 biomolecules-14-01159-f006:**
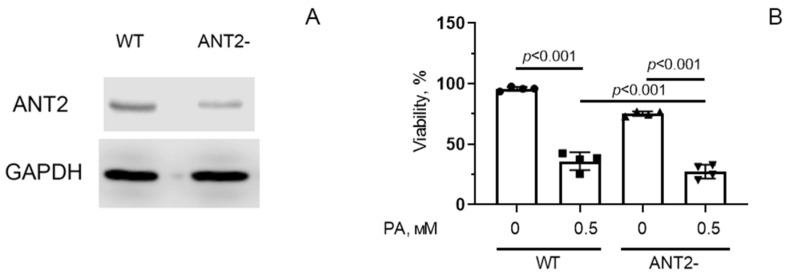
The amount of ANT2 protein in HEK293T cells with normal (WT) and decreased (ANT2-) expression of ANT2 (**A**). Survival of HEK293T cells with normal and reduced expression of ANT2 under conditions of palmitate PA-induced lipotoxicity (0.5 mM PA/fatty acid-free BSA complex solution for 6 days) (**B**). Conditions: 0—BSA solution, 0.5 PA—0.5 mM PA/BSA complex solution. Data represent the mean ± SD from four independent experiments, including at least 25 fields of view. Original images of (**A**) can be found in Appendix A.

**Figure 7 biomolecules-14-01159-f007:**
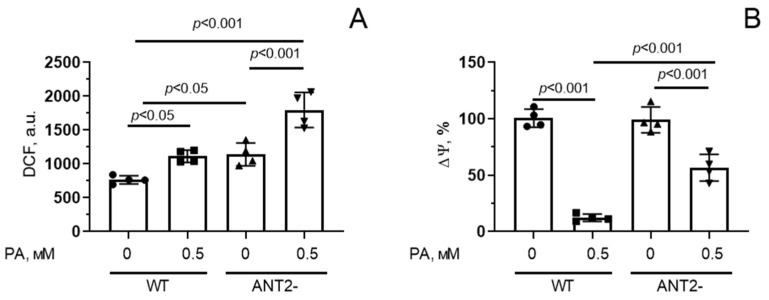
Changes in production of reactive oxygen species (**A**) and mitochondrial membrane potential (Δψ) (**B**) in HEK293T cells with normal (WT) and reduced (ANT2-) expression of ANT2 under conditions of PA-induced lipotoxicity (0.5 mM PA/fatty acid-free BSA complex solution for 48 h). Conditions: 0—BSA solution, 0.5 PA—0.5 mM palmitate/BSA complex solution. Data represent the mean ± SD from four independent experiments, including at least 25 fields of view.

**Figure 8 biomolecules-14-01159-f008:**
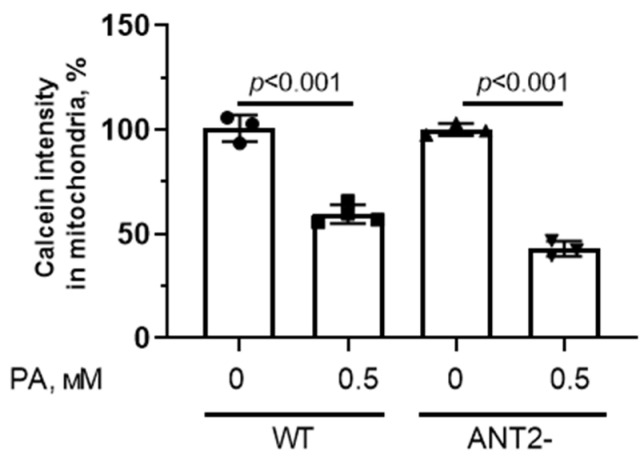
MPT pore opening in HEK293T cells with normal (WT) and reduced expression of ANT2 (ANT2-). Intensity of calcein fluorescence in HEK293T cell mitochondria from four experimental groups. Conditions: 0—BSA solution, PA—0.5 mM palmitate/BSA complex solution. Data represent the mean ± SD from 3–4 independent experiments, including at least 25 fields of view.

**Table 1 biomolecules-14-01159-t001:** Nucleotide sequences and characteristics of gRNA protospacers and primers, used for genome editing and amplification of the target DNA site.

Name	Sequence	Amplicon Length, bp	T_annealing_, °C
SLC25A5-gRNA1	TGCAAAGTAGAGCCAAAACT	-	-
SLC25A5-gRNA2	TGGCATCGGGTGGTGCCGCA		
SLC25A5-test-f	AGGGTCTGAAGGTCACACGGGT	822	58
SLC25A5-test-r	GGACTGTTAGGTTGGTTGGTACAATGC		

## Data Availability

The data presented in this study are available upon request from the corresponding author.

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
