# Peer review of "ANT-Mediated Inhibition of the Permeability Transition Pore Alleviates Palmitate-Induced Mitochondrial Dysfunction and Lipotoxicity"

_biomolecules, 2024, doi:10.3390/biom14091159_

Round 1
Reviewer 1 Report
Comments and Suggestions for Authors
The authors aimed to evaluate the preventive effects of BA and Catr on mitochondrial dysfunction induced by lipotoxicity in a primary culture of mouse pulmonary vascular endothelium. They also investigated the impact of ANT2 silencing on the development of mitochondrial dysfunction, oxidative stress, and survival of stably transfected HEK293T cells exposed to high PA concentrations.
The manuscript addresses an important issue in understanding the mechanisms involved in lipotoxicity conditions. Although the authors' concepts and observations are interesting, some points need to be clarified and discussed further. I recommend accepting the manuscript after the major changes are addressed.
Some sentences in the manuscript are difficult to read because they are too long. The entire manuscript should be revised to address this issue. Additionally, some references are missing in certain paragraphs. For example, in line 57: "Indeed, in vitro and in vivo models of diabetes and its related complications (hyperglycemia, lipotoxicity, etc.) have demonstrated that mitochondrial dysfunction is accompanied by increased production of ROS by the electron transport chain, disruption of the process of oxidative phosphorylation, a drop in the mitochondrial membrane potential, and the opening of the calcium-dependent non-selective membrane pore called the mitochondrial permeability transition (MPT) pore."
Several methodological details need clarification:
- How were the control cells treated? What about the vehicle treatments with BSA and DMSO?
- How long were the cells treated with BA and CA? Did this treatment occur before or after PA exposure?
Specific concerns regarding Figure 1:
- In Fig. 1A, why did the authors choose a concentration of 10 µM of CA instead of 5 µM, given that neither concentration induces a decrease in cell viability? Could the toxic effect observed with the co-treatment with PA be prevented?
- In Fig. 1B, why is there no control group? It is necessary to prove that under your conditions, PA induces a decrease in cell viability. How many experiments were performed in Fig. 1?
Line 299: "Since 0.75 mM palmitic acid caused a sharp decline in cell viability within 6 days, in the next series of experiments we simulated lipotoxicity over a 48-hour period." This statement is not demonstrated in the manuscript. If this is based on a previous study, that study should be cited.
There is a discrepancy in the results observed with DCF and MitoSox for PA+BA. For DCF, there is a reduction in ROS production, but it is not significant compared to PA. However, in the MitoSox assay, there is complete inhibition of ROS production. This discrepancy should be discussed.
BA and CA are modulators of ANT in the inner mitochondrial membrane, yet they have opposite effects in cells treated with PA. Why? What are the primary mechanisms of BA and CA that could justify these different results?
The graphs should represent individual levels for each experiment, perhaps using a scatter plot with bars. The number of experiments in each assay is very small (3 or 4), especially for primary cell cultures, which have significant variability. A scatter plot would allow the variability of each experiment to be observed.
In Figure 6B, the authors compared the control of WT and ANT-KO cells. It seems that the ablation of ANT2 by itself induces a reduction in cell viability. Why did you not ablate the expression of ANT2 in the primary cultures using siRNA? Doing so would clarify whether the effect is mediated by ANT. Can you discuss this?
There are several references that should be added to the manuscript and discussed in relation to your results ( Ortiz-Rodriguez, Acaz-Fonseca et al. 2019, Schmitt, Blanco et al. 2024).
Ortiz-Rodriguez, A., E. Acaz-Fonseca, P. Boya, M. A. Arevalo and L. M. Garcia-Segura (2019). "Lipotoxic Effects of Palmitic Acid on Astrocytes Are Associated with Autophagy Impairment." Mol Neurobiol 56(3): 1665-1680.
Schmitt, L. O., A. Blanco, S. V. Lima, G. Mancini, N. F. Mendes, A. Latini and J. M. Gaspar (2024). "Palmitate Compromises C6 Astrocytic Cell Viability and Mitochondrial Function." Metabolites 14(3).
Minor comments:
- What is VDAC1? The extended name should be added.
- The amount of protein used for immunoblotting should be specified in the Methods section.
Reviewer 2 Report
Comments and Suggestions for Authors
The paper entitled “ANT-mediated inhibition of the permeability transition pore underlies palmitate-induced mitochondrial dysfunction and lipotoxicity” prensented for publication in biomolecules by Natalia V. Belosludtseva et al. is of excellent quality and constitute a new source of valuable informations. So, It is worth to make it published but after some additive experimental work.
Major comment:
You use DCFH-DA for ROS measurement whereas the mitochondrially produced ROS are mostly superoxide anions that are easily measured with MitoSOX probes. This has to be done and the related comments added to the publication. The use of specific inhibitor will be welcome as proof of concept.
You say: Fig. 5 shows that under conditions of 48-h hyperlipidemia, the colocalization of mitochondria and lysosomes significantly increased, which may indicate increased mitochondrial elimination (as well as the induction of mitophagy). But you did not work on proving that the cells undergo a classical mitophagy.
It is also a pity that you did not use electron microscopy to prove that mitochondria could be find in autophagosomes.
Minor comments:
Please comment the possibility of a high toxicity of BA (not related with its action on the mitochondrial permeability transition.
For carboxy-atractyloside the abbreviation CAT will be better than Catr. Please adapt the text.
Round 2
Reviewer 1 Report
Comments and Suggestions for Authors
The authors properly addressed all the issues raised. I recommend to accept the manuscript.
Reviewer 2 Report
Comments and Suggestions for Authors
Since the authors have clearly responded to the reviewer demand. I don't see any reasons not to accept the present manuscript.